

# Dawn chorus interpretation differs when using songs or calls: the Dupont's Lark *Chersophilus duponti* case

Cristian Pérez-Granados[1,3], Tomasz S. Osiejuk[2] and Germán M. López-Iborra[1]

[1] Ecology Department/Multidisciplinary Institute for Environmental Studies "Ramón Margalef", Universidad de Alicante, Alicante, Spain
[2] Department of Behavioural Ecology, Institute of Environmental Biology, Faculty of Biology, Adam Mickiewicz University of Poznan, Poznan, Poland
[3] Current affiiation: Terrestrial Ecology Group (TEG-UAM), Department of Ecology, Universidad Autónoma de Madrid, Madrid, Spain

## ABSTRACT

**Background**. Territorial songbirds vocalise intensively before sunrise and then decrease their vocal activity. This creates a communication network that disseminates essential information for both males and females. The function of dawn chorus in birds has been frequently interpreted according to seasonal variation of singing as the breeding season advances, but potential differences in seasonal variation of song and calls for the same species have not been taken into account.

**Methods**. We chose Dupont's Lark as a model species to study whether the seasonal pattern of dawn chorus differs between singing and calling activity, because in this species most daily songs and calls are uttered at dawn. We registered vocal activity of Dupont's Lark before and around dawn in three different populations, through repeated sampling over the entire breeding season of two consecutive years.

**Results**. We found that dawn singing parameters remained constant or presented an increasing trend while dawn calling activity decreased as breeding season advanced. We also found different daily patterns for singing and calling, with birds calling mostly during the first 30 minutes of dawn choruses and singing peaking afterwards.

**Discussion**. The different time patterns of songs and calls may indicate that they serve diverse functions at dawn in the Dupont's Lark. Relaxation of dawn calling activity after the first month of the breeding season would suggest that dawn calling may be mainly related to mate attraction, while constant dawn singing throughout the breeding period would support a relationship of dawn singing to territorial defence. Our study highlights that the type of vocalisation used is an important factor to consider in further research on dawn choruses, since results may differ depending on whether calls or songs are analysed.

## INTRODUCTION

Dawn chorus occurs when birds, usually males, begin to sing before sunrise and then cease or decrease song activity for the rest of the day (*Mace, 1987*; *Catchpole & Slater, 2008*, but see *Garamszegi et al., 2006* and *Webb et al., 2016* for female singing). It is a phenomenon

Corresponding author
Cristian Pérez-Granados,
cristian.perez@ua.es

found in many bird species and has been studied since the late 19th and early 20th centuries (e.g., *Wright, 1912*; *Allen, 1913*). Previous works have suggested that the dawn chorus may be regarded as a reliable signal of male quality (e.g., *Grava, Grava & Otter, 2009*; *Marini et al., 2017*). To understand dawn chorus in territorial birds, studies should account for: (1) the reason of the concentration of vocal activity at dawn, which has been explained by female behaviour, conditions for sound transmission and low foraging success at dawn, among others (*Kacelnik & Krebs, 1983*; *Kunc, Amrhein & Naguib, 2005*; *Naguib et al., 2016*), and (2) the function of the vocalisations uttered at dawn, which has been related to mate attraction (*Eriksson & Wallin, 1986*; *Poesel et al., 2006*; *Murphy et al., 2008*) and to territorial defence and mate guarding (*Møller, 1988*; *Møller, 1991*; *Amrhein & Erne, 2006*), although the two functions can be complementary (*Møller, 1991*; *Slagsvold, Dale & Sætre, 1994*).

Despite the prevalence of dawn-singing in birds and the numerous studies focused on that topic, its origin, evolution and causes remain unclear and seem to be context-dependent, differing among species, populations and studies (e.g., *Staicer, Spector & Horn, 1996*; *Catchpole & Slater, 2008*; *Zhang, Celis-Murillo & Ward, 2016*). This could be explained in part because researchers have indistinctly analysed or have even pooled calls and songs (*Poesel, Foerster & Kempenaers, 2001*; *Grava et al., 2013*; *Stanley et al., 2016*; *Lee, MacGregor-Fors & Yeh, 2017*) to elucidate dawn chorus function (but see *LaZerte, Otter & Slabbekoorn, 2017*). Nonetheless, the bulk of dawn chorus studies have been focused on singing activity (e.g., *Gil, Graves & Slater, 1999*; *Amrhein, Korner & Naguib, 2002*; *Liu, 2004*). Although the distinction between calls and songs is not always clear-cut, in practice a given vocalisation can usually be readily classified as either a song or a call, and these classifications are commonly used in the birdsong literature (*Zann, 1990*; *Wang et al., 2012*). In oscines, hummingbirds and parrots, songs are defined as long, loud, complex, and learned spontaneous vocalisations whose function at dawn has usually been related to mate attraction, but also to compete with members of the same sex (*Slagsvold, Dale & Sætre, 1994*; *Catchpole & Slater, 2008*). Calls are usually described as shorter, simpler, less spontaneous and genetically determined vocalisations uttered throughout the year and that can be produced for a variety of reasons including maintaining group contact and signalling about food or danger, such as in cases of fight, threat or alarm (*Marler, 2004*; *Catchpole & Slater, 2008*). The use of calls in dawn chorus has been little studied (but see *LaZerte, Otter & Slabbekoorn, 2017*) and is not well understood. In some species, it has been suggested that calls are targeted to the current female (*McCallum, Grundel & Dahlsten, 1999*; *Stowell, Gill & Clayton, 2016*). Thus, as the origin and evolution of the two types of vocalisations differ, the interpretation of dawn chorus function could vary according to the type considered.

In this study, we selected the Dupont's Lark (*Chersophilus duponti*) as a model to explore potential differences in the interpretation of dawn chorus when analysing songs or calls. We chose this species because it is a diurnal passerine with high vocal activity concentrated at dawn and because it utters a large number of both types of vocalisations during dawn choruses. There are several studies published about the song and the territorial call of this species, which provide good background for deeper analysis (*Laiolo & Tella, 2005*; *Laiolo & Tella, 2006*; *Laiolo, 2008*; *Laiolo et al., 2008*; *Pérez-Granados, Osiejuk & López-Iborra, 2016*).

All of these studies have assumed that only males sing and produce territorial calls, since females do not appear to utter long range vocalisations (*Laiolo et al., 2007*). Occasionally, Dupont's Lark males also produce warning calls (to repel intruders from their territories, *Laiolo et al., 2005*) and both sexes may utter distress calls, an ultimate alarm signal given by individuals that are in predator risk (*Laiolo et al., 2005*, C Pérez-Granados, pers. obs., 2018). Although some female larks are known to sing, such as the Eurasian Lark (*Alauda arvensis*) or the Woodlark (*Lullula arborea*) (*Garamszegi et al., 2006*; *Odom & Benedict, 2018*), there is no evidence of Dupont's Lark females singing or uttering territorial or warning calls (*Laiolo & Tella, 2005*; *Garamszegi et al., 2006*; *Laiolo, 2008*; *Laiolo et al., 2005*; *Laiolo et al., 2007*; *Laiolo et al., 2008*; *Barrero et al., 2017*; C Pérez-Granados, pers. obs., 2011–2012), despite the large number of bioacoustic studies focussed on the species and the fact that hundreds of Dupont's Larks have been ringed and later resighted singing. Therefore, in this study we considered that all monitored birds uttering songs or territorial calls were males. We are aware that females are able to utter distress calls in the nest and during handling (C Pérez-Granados, pers. obs., 2018), but this type of vocalisation, as well as warning calls, were not included in our study and we will refer to territorial calls simply as calls hereinafter.

In this paper, we aimed to investigate if singing and calling activity at dawn in the Dupont's Lark have different seasonal patterns and to explore the consequences of using songs or calls when interpreting dawn chorus function. For this purpose, we monitored the dawn singing and calling activity of the Dupont's Lark during the breeding season in two consecutive years. We hypothesised that if songs and calls have different functions, their seasonal pattern throughout the breeding period would differ (*Celis-Murillo et al., 2016*). Vocalisation functions cannot be demonstrated without experimental or more detailed observational analyses. However, we assumed that the relationship between function and seasonal patterns of vocalisations in the Dupont's Lark is similar to that described for other European passerines. Therefore, we expected that Dupont's Lark singing or calling activity at dawn would decrease after pairing if they are related to attracting females at that time, as described for other passerine species (e.g., *Gil, Graves & Slater, 1999*; *Amrhein, Korner & Naguib, 2002*; *Liu, 2004*; *Celis-Murillo et al., 2016*), but would remain constant if they are related to territorial defence (e.g., *Olinkiewicz & Osiejuk, 2003*; *Kunc, Amrhein & Naguib, 2005*; *Amrhein & Erne, 2006*; *Liu & Kroodsma, 2007*).

# MATERIAL AND METHODS

## Study area

We conducted a field study from late March to early June in 2013 and 2014 in the three largest Dupont's Lark populations in Valencia province (eastern Spain, *Pérez-Granados & López-Iborra, 2013*), where patch size and number of territorial males ranged between 193–200 ha and 10–19 males, respectively. Study sites were located at 1,000 m a.s.l. and the vegetation was a shrub-steppe dominated by low shrub species (*Thymus* spp., *Genista scorpius* and *Rosmarinus officinalis*). In all sites, there were some dispersed trees, especially pines *Pinus* spp. and junipers (*Junipers oxycedrus* and *J. communis*). More information on the study area can be found in *Pérez-Granados, López-Iborra & Seoane (2017a)*.

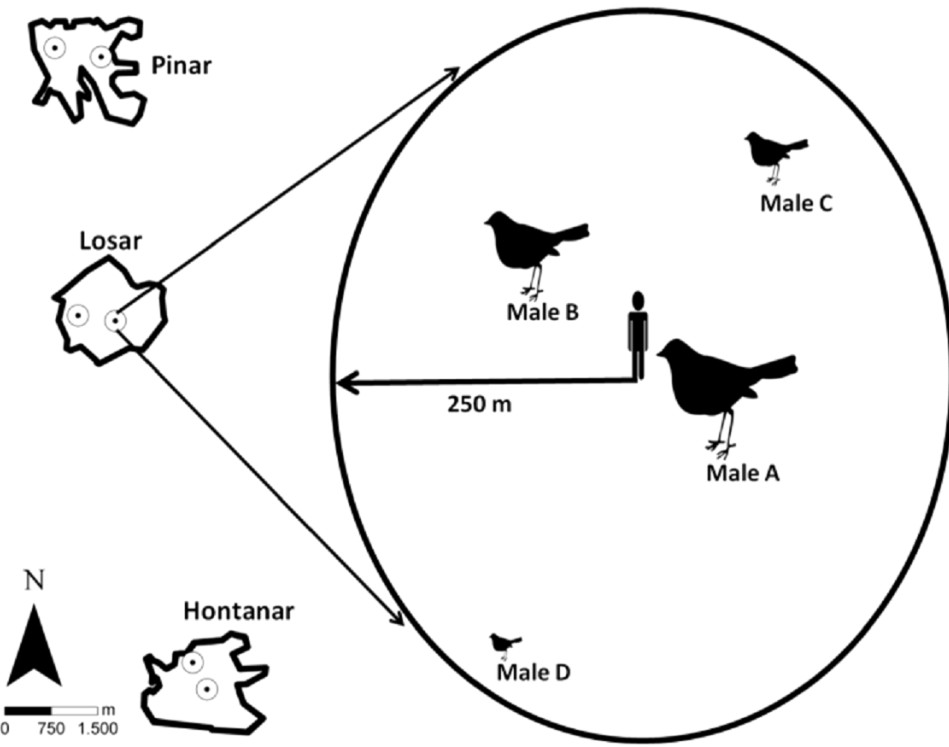

**Figure 1** **Location of the sampling stations in the three studied sites, and sampling station procedure.** Song activity was registered individually for each Dupont's Lark male, according to their different distance and location to observer.

## Data collection

We registered dawn singing and calling activity of Dupont's Lark males in three sites over two consecutive breeding seasons (years). Acoustic activity was measured through repeated sampling over the entire breeding season from one fixed counting station per site and year. Counting station location remained the same within a year but were separated by at least 500 m among years (Fig. 1) in order to monitor two different areas within each site. This is also expected to favour the monitoring of different males, since in the study area adult Dupont's Larks have been reported to move on average 100 m between breeding seasons (*Pérez-Granados & López-Iborra, 2015*). Likewise, only about half of the adult males survive between successive years (*Laiolo et al., 2008*), which would further decrease the probability of detecting the same male in the two study years. Songs and calls were assigned to a specific male based on differing direction and distance to the observer (Fig. 1). We only considered birds located within a 250 m radius from a station location that did not change their location during the survey (estimated displacement less than 50 m) to facilitate recognition. Distance from observer to vocalising bird was determined upon acoustic clues and based on previous experience counting the species in the study area (e.g., *Pérez-Granados & López-Iborra, 2013*; *Pérez-Granados, López-Iborra & Seoane, 2017a*). This wide radius was considered given that the species' songs may be heard from

long distances (*Laiolo et al., 2007*; *Vögeli et al., 2010*). We monitored vocal activity of up to a maximum of four males per station and night, with a mean (±SD) number of 3.5 ± 0.5 males monitored per station/night.

Surveys were labelled as ''1, 2, 3, 4, 5 or 6'', following the order in which they were conducted. In both years, the first survey was carried out between the 26th and 31st of March, and successive surveys were carried out fortnightly from then until 14–16 of June. This period corresponds to most of the breeding period of the species (*Herranz et al., 1994*; *Pérez-Granados et al., 2017b*). The first two surveys (between late March and early April) correspond to the first stage of the breeding time in the study area, when most females are building their nests and laying first clutches. The third and fourth surveys (late April–early May) were carried out at the peak of the breeding activity of the species (*Herranz et al., 1994*; *Pérez-Granados et al., 2017b*). During this time of the season, breeding pairs are rearing their first clutch or, in some cases, second or replacement clutches. The fifth and sixth surveys (late May–early June) were carried out at the end of the breeding season, a time when some breeding pairs may have finished their breeding period and a relaxation in territory defence may occur (*Pérez-Granados et al., 2017b*).

Dawn acoustic activity was always monitored by the same researcher (CPG), who stood at the station from 100-min before sunrise to ten minutes after dawn. The observer stood for 5 min in silence at each station before the study began. Daily times of sunrise at the geographic location of the study area were obtained from the Spanish Ministry of Development (http://astronomia.ign.es/web/guest/hora-salidas-y-puestas-de-sol). We divided the continuous survey into 5-min intervals and registered the number of songs and calls uttered per male per 5-min period. For each interval, we calculated song/call output as the total number of songs/calls uttered per male. We also monitored song/call start and end time (5-min interval in which the first and the last song or call were produced by any male each night) and performance time, measured as the total number of 5-min intervals that each bird was vocally active each night.

The song of the Dupont's Lark in the study area includes a mean of five discrete song types with no differences in male repertoire size among studied sites (*Pérez-Granados, Osiejuk & López-Iborra, 2016*). Song types are largely shared and repeated in the same order by neighbouring males and end with a common sequence of the species' so-called 'whee-ur-wheeee' (*Cramp, 1988*). We considered the whole sequence of repeated song types as a unique song to facilitate data collection and used the definite 'whee-ur-wheeee' to define the boundary between one song and another. A sonogram of a typical sequence recorded in the study area can be found in Fig. 2 and a song recorded in the study area may be found in Pérez-Granados (XC216990, on http://www.xeno-canto.org). Sonograms of the commonest song types in the three studied sites can be found in *Pérez-Granados, Osiejuk & López-Iborra (2016)*. Calls consist of discrete and short whistles (a complete description and sonograms can be found in *Laiolo et al., 2007*).

Surveys were conducted only during windless dawns with good weather conditions (cloud cover <10% and no rain) to avoid potential biases in bird behaviour. Dawn singing activity of the Dupont's Lark varies with moon phase (authors' own data). Therefore, surveys were carried out on the two days following or preceding full or new moons, to

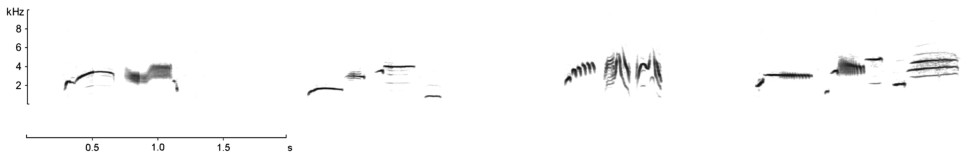

**Figure 2**  Sonogram of a typical Dupont's Lark song in the study area.

control for the potential effect of moon phase in the analyses. Moon phases were obtained from the Spanish Ministry of Development (http://astronomia.ign.es/web/guest/agenda-astronomica).

## Statistical analyses

All statistical analyses were performed with R v. 3.4.1 (*R Development Core Team, 2014*) and all results are expressed as mean ± SE. To analyse the variation of dawn singing or calling activity across the breeding season, we fitted independent Linear Mixed Models for songs and calls. Start time, end time, performance time or total night output were used as response variables. Survey as a continuous variable ("1/2/3/4/5/6") was included in models as a fixed effect and site ("Hontanar/Losar/Pinar"), moon phase ("Full/New"), and year ("2013/2014") as random effects to control for variation owing to site, lunar cycle and inter-annual variations. Linear Mixed Models were fitted using the "lmer" function in the R package "lme4" (*Bates et al., 2015*). The degrees of freedom were calculated using the Kenward-Roger method (*Kenward & Roger, 1997*). We present models with *p*-value significance for fixed effects calculated from type III *F*-tests using the lmerTest package (*Kuznetsova, Brockhoff & Christensen, 2014*).

We also calculated for each survey (pooling sites and years) the Pearson correlation between the total number of songs and calls uttered per male to ascertain whether there was a trade-off between the two types of vocalisations. To evaluate similarity between surveys in the temporal pattern of vocalisations in relation to dawn, we calculated a Pearson correlation matrix between surveys for each type of vocalisation (mean number of songs or calls uttered per male per 5-min period). Each correlation matrix was used as the similarity matrix in a cluster analysis (Unweighted pair-group method with arithmetic averages, UPGMA method, *Sneath & Sokal, 1973*) aimed at identifying groups of surveys with similar dawn activity patterns for songs and calls.

## RESULTS

The only characteristic of singing activity related to the advance of the breeding season was singing output, since total number of songs uttered per male and night significantly increased in later surveys, while singing performance and singing start or ending time did not show a significant trend through surveys (Table 1, Fig. 3). On the contrary, calling activity was strongly related to the advance of the breeding season since calling start time, calling performance period and calling output differed significantly among surveys, while

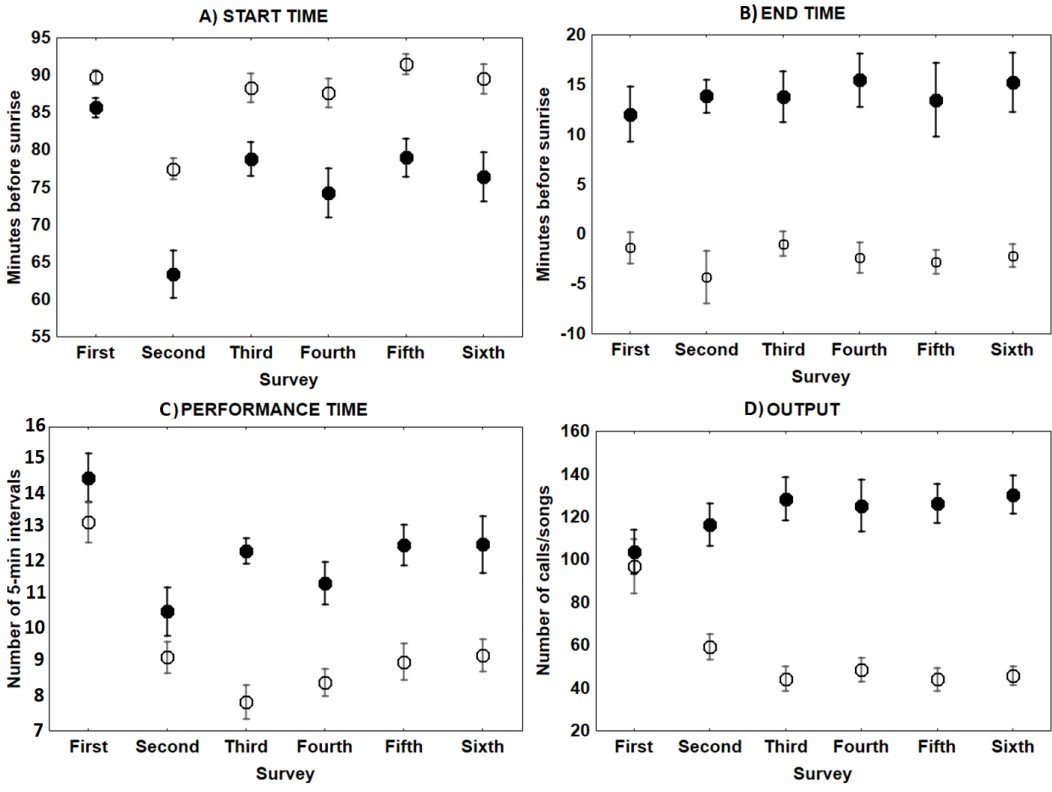

**Figure 3** Mean ± SE singing and calling activity at dawn of Dupont's Lark during the breeding cycle of the years 2013 and 2014 at Rincón de Ademuz (eastern Spain). Vocal activity was monitored in six surveys, performed fortnightly from late March (First Survey) to early June (Sixth Survey). Start time (A, minutes before sunrise when the first song or call was uttered), End time (B, minutes before or after sunrise when last song or call was produced), Performance time (C, total of 5-min intervals that each bird was active) and Output (D, total number of songs or calls uttered per male), are expressed for dawn singing activity (filled circles) and dawn calling activity (empty circles).

calling end time did not (Table 1). Males initiated calling activity earlier at the end of the breeding season (Fig. 3), despite the fact that they produced significantly more calls and called for more time during the first surveys of the breeding season in comparison to those at the end of the breeding time (Fig. 3 and see Table S1). We found a significant negative correlation between the number of songs and the number of calls uttered per male each night during the first survey, which was still significant but less intense in the second survey (Fig. 4 and see Table S2). We found no correlation in later surveys, thus showing a decreasing intensity of the correlation between both types of vocalisations as breeding season advanced (Table S2).

For singing and calling activity, classification analyses identified two periods with different vocalisation distribution before dawn that included in one group the first two surveys and the remaining four surveys in another (Fig. 5). Time patterns of vocal activity before dawn also differed between vocalisation types (Fig. 6). Singing activity showed a symmetrical, bell-shaped response curve centred in the vocalisation interval, while calling

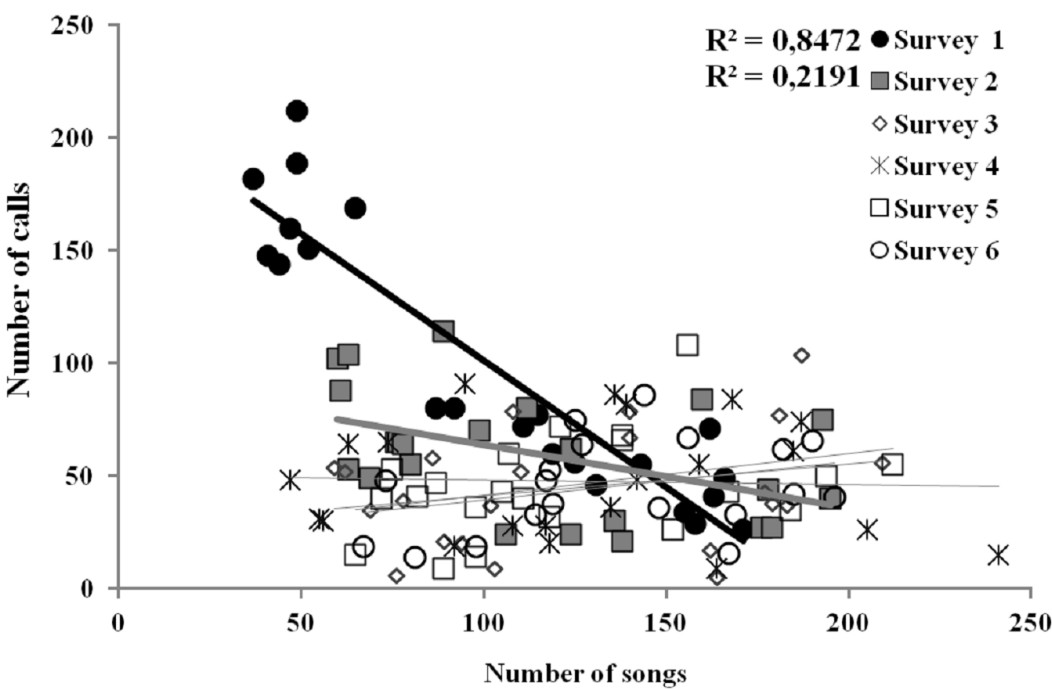

**Figure 4** **Linear relationships between the number of songs and calls uttered per each Dupont's Lark male and survey (data of 2013 and 2014 pooled) at Rincón de Ademuz (eastern Spain).** Vocal activity was monitored in six surveys, performed fortnightly from late March (First Survey) to early June (Sixth Survey). Surveys with significant Pearson's correlations are represented with filled symbols and their $R^2$ value can be found in legend (see complete analyses in Table S2).

**Table 1** **Test of significance of the survey period as fixed effect in the GLMMs for different response variables describing singing and calling behaviour of the Dupont's Lark at dawn.**

| | Songs | | | Calls | | |
|---|---|---|---|---|---|---|
| Response variable | Sum Sq | F | Pr(>F) | Sum Sq | F | Pr(>F) |
| Start time | 72.65 | 1.32 | 0.252 | 902.46 | 45.83 | <0.001 |
| End time | 102.92 | 0.90 | 0.343 | 1.65 | 0.034 | 0.854 |
| Performance time | 1.38 | 0.22 | 0.639 | 96.28 | 17.58 | <0.001 |
| Output | 6705.3 | 4.48 | 0.036 | 25704 | 21.70 | <0.001 |

activity peaked between 85–75 min before dawn and decreased from that time. Time patterns of both vocalisation types were advanced with respect to dawn in the second group of surveys and were remarkably similar within each group. In addition, calling pattern during the last phase of the breeding season presented a bimodal pattern, with an almost total absence of calls between 30 and 45 min before dawn, which was lacking in the first two surveys (Fig. 6).

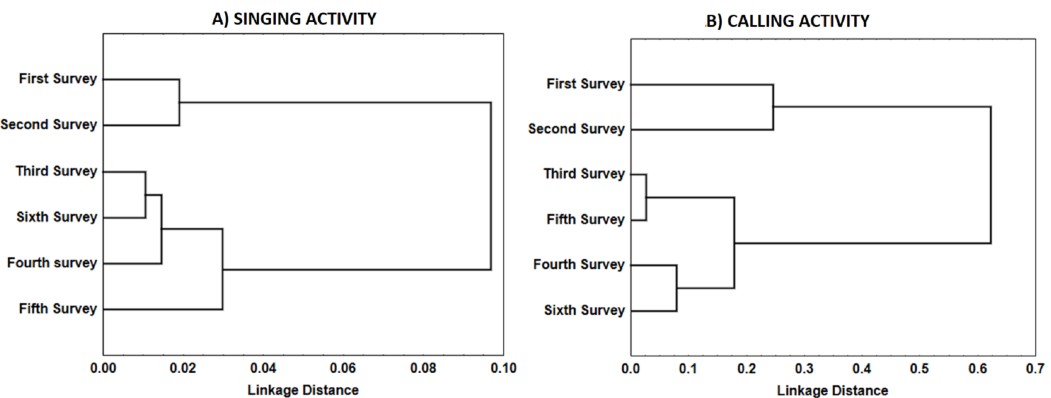

**Figure 5** **Similarity in the distribution of each type of vocalisations (A: songs; B: calls) during Dupont's Lark dawn choruses.** Dendrogram resulted from hierarchical cluster analyses using complete linkage. Pearson's correlation between surveys in the mean number of songs or calls uttered per male per 5-min period was used as similarity measure. Correlations were converted to distances computed as 1.0 minus Pearson r.

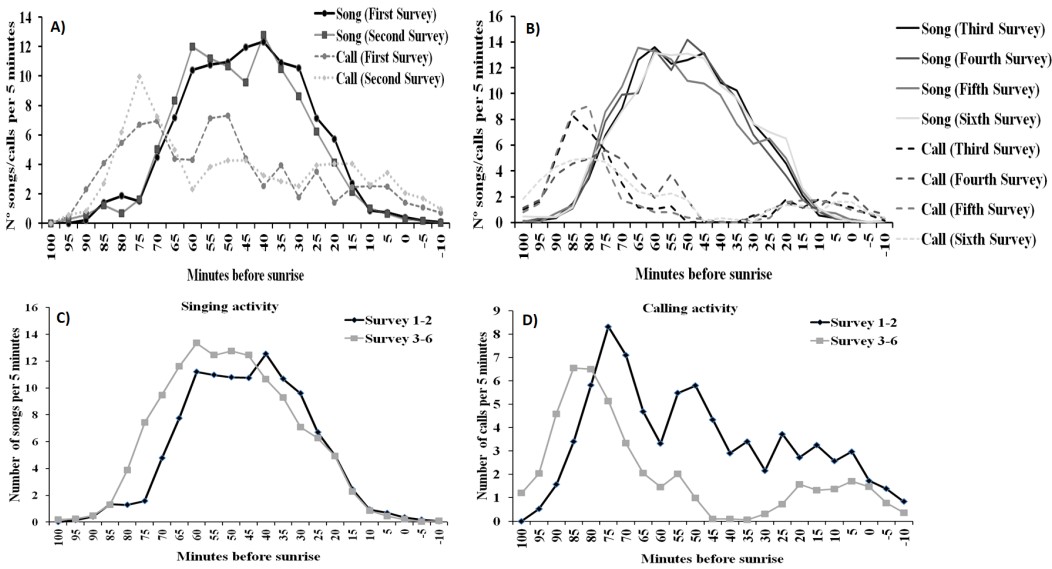

**Figure 6** **Dawn vocal activity pattern of Dupont's Lark during the breeding cycle of the years 2013 and 2014 at Rincón de Ademuz (eastern Spain).** Vocal activity was monitored in six surveys, performed fortnightly from late March (First Survey) to early June (Sixth Survey). Vocal activity is expressed as the mean number of songs or calls uttered per male per 5-min interval. Mean singing (filled line) and calling (dotted line) activity in the first (A) and the second (B) periods identified by cluster analysis (see Fig. 4). Mean daily pattern of singing (C) and calling (D) activity for the surveys included in the periods defined in Fig. 4.

## DISCUSSION

To our knowledge, this is the first study to analyse seasonal patterns of dawn chorus of a passerine using songs and calls with data collected at the same time (but see *LaZerte, Otter & Slabbekoorn, 2017*). This is also the first study to quantitatively describe the dawn chorus of the Dupont's Lark, an endangered species. We found that dawn seasonal patterns of singing and calling behaviour differed significantly: dawn singing output increased with the advance of the breeding period while dawn calling activity decreased and started earlier as the breeding season advanced. The different seasonal patterns of calls and songs suggest that they serve diverse functions at dawn (*Celis-Murillo et al., 2016*) and therefore, the use of one or the other vocalisation could lead to different conclusions about dawn chorus function in the studied species.

In several passerine species, a pattern of greater dawn singing activity at the beginning of the breeding period has been described, followed by a relaxation after pair formation, in agreement with the hypothesis that dawn song plays a role in mate attraction (e.g., *Gil, Graves & Slater, 1999*; *Amrhein, Korner & Naguib, 2002*; *Liu, 2004*). Conversely, most of the dawn singing activity parameters analysed in the Dupont's Lark remained constant throughout the breeding season and even song output showed a seasonal increasing trend over time. This temporal pattern does not agree with the patterns expected if the function of songs was to attract females. This suggests that dawn song could play a role in territorial defence in the studied species due to constant dawn singing activity over time, as has been proposed in other passerines (*Olinkiewicz & Osiejuk, 2003*; *Amrhein, Kunc & Naguib, 2004*; *Liu, 2004*; *Kunc, Amrhein & Naguib, 2005*). Previous studies have assumed that dawn singing behaviour in passerines can be used in both male and female interaction contexts, with their functions difficult to ascertain (*Slagsvold, Dale & Sætre, 1994*; *Olinkiewicz & Osiejuk, 2003*; *Zhang, Celis-Murillo & Ward, 2016*). Both functions have been assigned especially to continuous and loud singing species at dawn, such as the Dupont's Lark, which are able to repel intruders while attracting potential mates or extra-pair mates (*Møller, 1991*; *Naguib et al., 2011*).

Contrary to dawn song, most of the calling activity parameters analysed changed as the breeding season advanced. The number of calls and the amount of time dedicated to calling decreased as breeding season progressed, especially after the first two surveys. *Laiolo et al. (2008)* also found that Dupont's Lark calling activity peaked in the first months of the breeding period. This seasonal pattern is compatible with the pattern expected if this vocalisation serves to attract mates, as has been proposed for other passerine birds (e.g., *McCallum, Grundel & Dahlsten, 1999*; *Gil, Graves & Slater, 1999*; *Amrhein, Korner & Naguib, 2002*; *Liu, 2004*). In multi-brooded species, vocalisations that attract females are maintained, albeit with reduced intensity. This can be explained as a male response to maintain their mates and quickly mate between successive clutches or if the nest is predated or fails (*Mace, 1986*; *Pärt, 1991*). The same explanation could be valid for the Dupont's Lark, as this species has a long breeding season with up to three clutches per year (early March–early July) and high nest predation rates (*Herranz et al., 1994*; *Pérez-Granados et al., 2017b*).

Song start time did not vary over time while males tended to initiate calling activity earlier with the advance of the breeding season. If calls were used for mate attraction, this result may be related to acquiring extra-pair copulations, which usually occur before dawn, a phenomenon well established among larks (*Sánchez et al., 2004*; *Hutchinson & Griffith, 2008*). Previous studies on passerines have found that males who began their daily dawn chorus the earliest were the most successful in obtaining extra-pair copulations (*Poesel et al., 2006*; *Dolan et al., 2007*). Time relative to dawn of the last call or song uttered did not differ throughout the breeding season, likely because the end of dawn choruses is related to increasing light intensity as sunrise approaches (e.g., *Da Silva et al., 2014*; *York, Young & Radford, 2014*). However, this result may be influenced by the methodology employed, since on some occasions males continued calling after surveys ended, which may have influenced our results.

The number of songs and calls uttered per bird each night were strong and negatively related in the first survey, but this relationship weakened with the advance of the breeding season and was non-existent from the third survey onwards. This pattern could be the consequence of a trade-off between territorial defence and mate attraction (*Hasselquist & Bensch, 1991*; *Slagsvold, Dale & Sætre, 1994*), because males may utilise different acoustic strategies (calling or singing) when attracting mates or defending territories (*Zhang, Celis-Murillo & Ward, 2016*). This trade-off may be enhanced in species, like Dupont's Lark, that concentrate vocal activity in short periods in the daily cycle. A larger proportion of unpaired males at the first stage of the breeding season would make this trade-off more apparent at that time, as these males would invest more time in calling for attracting females at the cost of reducing song output, while after pairing more efforts would be devoted to singing.

We found different daily patterns at dawn for singing and calling activity. In general, Dupont's Lark uttered calls mostly during the first 30 min of dawn choruses with songs being the commonest vocalisation onwards. The mean daily pattern of singing and calling activity were similar along the breeding season, but that pattern was more advanced in the later surveys, with birds singing and calling earlier in time. This definite daily pattern together with seasonal differences found between vocalisation types support our assumption that calls and songs may serve diverse functions in the studied species, as has been proposed for other passerines (*Celis-Murillo et al., 2016*). According to hypothetical functions assumed for dawn singing and dawn calling in the studied species, our results suggest that males may dedicate more time to mate attraction at the beginning of dawn choruses, under low-light conditions, and to territorial defence from that moment on. These different patterns may also reflect a trade-off between singing and calling within each period before dawn, as the peak in singing is associated with a decrease in calling. The temporal pattern of calling becomes bimodal from the third survey onwards, as calling activity ceases completely between 45 and 35 min before dawn, just after the peak in signing. Song output increases as season advances, and particularly after the second survey, and thus cessation of calling could be due to increasing efforts in singing.

## CONCLUSIONS

In this study, we found evidence that dawn singing and calling activity in Dupont's Lark present different temporal patterns of activity, both along the breeding season and in the period before dawn, when most of the vocal activity of this species occurs. These different patterns suggest that calls and songs in the Dupont's Lark may have functions different to those expected. However, experimental studies are needed to obtain conclusive results about the concrete function of each vocalisation type in the studied species. To unravel the functions of these vocalisations, the next step should be to develop experimental studies testing responses of males and females to songs and calls that would provide additional insight into dawn chorus signalling functions in the Dupont's Lark. Nevertheless, our results highlight that the type of vocalisation used is an important factor to consider when seasonal changes of vocal activity are studied, and thus previous interpretations made for birds' dawn choruses may need to be re-evaluated in light of the type of vocalisation analysed. We encourage researchers to clearly describe the type of vocalisation used in future studies.

## ACKNOWLEDGEMENTS

We are grateful to three anonymous reviewers whose comments helped to improve the manuscript. We wish to thank the Servicio de Vida Silvestre of Conselleria d'Infraestructures, Territori i Medi Ambient (Generalitat Valenciana) and Juan Jiménez for supporting our work. We thank Sarah Young for help with the written English.

### Funding

This research was supported by the project "Estudio aplicado a la conservación de la conservación de las poblaciones de alondra ricotí (Chersophilus duponti) en el entorno del municipio de Vallanca" funded by 'Levantina y Asociado de Minerales, S.A.''. The funders had no role in study design, data collection and analysis, decision to publish, or preparation of the manuscript.

### Grant Disclosures

The following grant information was disclosed by the authors:
Levantina y Asociado de Minerales, S.A.

### Competing Interests

The authors declare there are no competing interests.

### Author Contributions

- Cristian Pérez-Granados conceived and designed the experiments, performed the experiments, analyzed the data, contributed reagents/materials/analysis tools, prepared figures and/or tables, authored or reviewed drafts of the paper, approved the final draft.
- Tomasz S. Osiejuk prepared figures and/or tables, materials/analysis tools, authored or reviewed drafts of the paper, approved the final draft.
- Germán M. López-Iborra conceived and designed the experiments, analyzed the data, contributed reagents/materials/analysis tools, prepared figures and/or tables, authored or reviewed drafts of the paper, approved the final draft.

## Data Availability

Pérez Granados, Cristians (2018): Dawn chorus Dupont's lark. figshare. Fileset. https://doi.org/10.6084/m9.figshare.5924641.v1.

## Supplemental Information

Supplemental information for this article can be found online at http://dx.doi.org/10.7717/peerj.5241#supplemental-information.

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
