# Peer review of "Dawn chorus interpretation differs when using songs or calls: the Dupont’s Lark Chersophilus duponti case"

_PeerJ, doi:10.7717/peerj.5241_

## Round 0.1 · original submission · Major Revisions

· Academic Editor

Major Revisions

I have sent your paper to three reviewers, and two of them suggested that your article can perhaps be accepted after a major revision. Therefore, I invite you to resubmit your manuscript after addressing ALL reviewer's comments.

The second reviewer recommended that the new version of the manuscript needs to be reviewed by a native English speaker to avoid grammar errors as well as improve the style and the flow of the arguments.

The third referee has critical requests that you should address, including a better way to report the statistical analyses.

When resubmitting your manuscript, please carefully consider all points mentioned in the reviewers' comments, explain every change made, and provide proper rebuttals for any remarks not addressed.

Reviewer 1 ·

Basic reporting

Very good, clear and well structured. Almost self-contained (but see general comments)

Experimental design

Good

Validity of the findings

Very good

Additional comments

This paper is a nicely written and clear observational study of the different patterns of calling and singing in a selected bird species.

I am a researcher into animal vocalisation but not an ornithologist; an ornithologist may have other things to say about the work.

The paper shows valuable quantitative evidence about the rates of singing and calling - for one species only, but with sufficient amounts of data and analysis to demonstrate multiple clear generalities about this species in this geographic location. Clear differences between singing and calling activity are demonstrated.

The discussion section provides a very good discussion of the findings, making connections to other findings and ideas in the field.

My requested changes, indexed by line number:

50: "when males start to sing" - should clarify that it is not only males in general. See http://femalebirdsong.org/

82: "it has been suggested that calls may be targeted to the current female" - note that this was quantitatively demonstrated in Stowell et al 2017 (for domesticated zebra finches) 10.1098/rsif.2016.0296

98: "We hypothesised that..." - this sentence is problematic. It conflates two things: the authors' hypothesis about the rates of vocal activity, and the assumption about the function of the vocalisations. Authors should split this sentence into two, to make it clear that the function is not a hypothesis but an assumption, and it is assumed but not tested in this work.

146: "we registered the number of songs" - how did the observer decide what is the boundary between one song and another? The authors should include a sonogram of XC216990 in this paper, annotated to show the reader how this would be counted. I do not know if the authors would consider it to be one song or many.

171: could the authors please explain (in reply) why "site" should be a random effect and not a fixed effect? It was fully within the authors' control and part of the design, and would seem to be a fixed effect.

182: "UPGMA method" - please provide a citation for this; it is not clear what it is.

246 "start time did not vary" - this should say "end time did not vary"

285 typo "signing"

293 typo "i the next step"

Method section generally: how were female vocalisations handled? Were any heard? How was it possible to discriminate them?

Results/discussion section generally: were the male calls observed to occur interactively, e.g. in alternation with female calls, or were there no apparent response relationships?

Reviewer 2 ·

Basic reporting

The literature cited seems adequate and the article structure is fine. The raw data are available and understandable. The paper is self-contained.

The English needs a lot of work. The authors struggle with when to use and not use definite and indefinite articles. For example, “the” should be inserted before “breeding season” in line 30, before “seasonal pattern” in line 33, before “dawn chorus” in line 53, and before “potential effect” in lines 162-163. The authors should insert “a” before “model species” in line 32, before “large number” in line 77, and before “station location” in line 123. Conversely, “the” could be omitted before “vocal activity” in line 56 and before “most of the breeding period” in line 131. These lists are probably not exhaustive. The authors also struggle with their choice of prepositions. For example, “focused on” rather than “focused in” (line 69), “from then” rather than “since then” (line 130) and “activity of up to” rather than “activity up to” in line 126. Verb tenses and number are also a recurring problem: in line 32 “chose” not “choose,” in line 32 “differs” not “differ,” and in line 237 “suggests” not “suggest.” A few sentences just do not make sense. For example, the third sentence in the abstract starts out: “Studies of the dawn chorus in birds have frequently interpreted their meaning in function of seasonal variation of singing as breeding season advances…” The referent for “their” is unclear as it is a plural pronoun that seems to refer to a singular subject (the dawn chorus). Beyond that, I am unable to make sense of the phrase “their meaning in function of seasonal variation…” Another problematic sentence is in lines 54-62. Overall, the manuscript would benefit from thorough editing for English usage.

Experimental design

Although not much can be inferred about function from seasonal patterns, a description of the patterns presumably would still meet the journal’s standards if it was competently done. The authors need to clarify a few points to allow readers to make this judgment. The authors state that data were taken only on males vocalizing within 250 meters of the observer although vocalizations could be heard from males over up to 1500 meters (lines 122-126). How was the distance to the vocalizing bird determined? Presumably this often had to be determined in the dark so that visual cues would not be available. Was there any check on these distance estimates? I’m also unclear on how a single observer simultaneously kept track of vocalizations from up to four males, with each producing up to about 20 songs and 20 calls per period (from the data files). Were the data spoken into a recording?

In the Data collection section of the Methods: please state early on the number of sites used per year, and the number of stations used per site.

GLMMs usually look for directional changes of a dependent variable with changes in the independent variables. Some of the results are written as if that were the case here (e.g. line 187 – “total number of songs … significantly increased”), but others results are written as if the analysis simply looked for variation without a directional trend (line 190 – “calling performance and calling output differed significantly among surveys”). Please clarify.

Validity of the findings

The authors describe the seasonal pattern of the dawn chorus of Dupont’s Lark, distinguishing between patterns in songs and calls, and then attempt to draw inferences about the function of the dawn chorus from their results. One problem with this approach is that it is in general quite difficult to discern much about the function of avian vocalizations from seasonal patterns, in large part because rates of vocalizations can change for reasons having little to do with their benefits (such as energy availability or opportunity costs). Perhaps the strongest inferences can be made for song when song rates decline drastically immediately upon mate acquisition, as occurs in some of the Acrocephalus warblers studied by Catchpole (e.g. Catchpole 1980 Behaviour 74:179). Catchpole, however, was following the reproductive cycles of the individual males whose song rates he was measuring, whereas in the present study the authors do not seem to know where the individuals they have studied are in terms of reproduction at different points in the season, or at least they do not tell the reader anything about this beyond rough seasonal averages. The authors remark (lines 132-134) that their first two surveys correspond to times when most females are building nests and laying their first clutches, which implies that most males were already mated by the first survey, so that the authors cannot ascribe changes in vocalization rates to pairing. Drawing inferences about the function of dawn chorus calls seems particularly difficult because the data do not differentiate between categories of calls (such as alarm calls versus contact calls). The suggestion that calls function in mate attraction (lines 237-238) seems especially dubious given the lack of evidence for a category of calls that might function in this way. I recommend in general backing away from conclusions about the functions of the vocalizations studied.

Additional comments

My two main recommendations are to (1) improve the English, perhaps by getting a native English speaker to edit the text, and (2) to water down any conclusions about the function of dawn chorus vocalizations based on these seasonal pattern data.

Reviewer 3 ·

Basic reporting

I see weak justification for the proposal of the study in the first place. Describing the differences in seasonal change between song and “calls” seems fairly trivial. Also, regarding the methodological application for monitoring of bird dawn chorus seems to me that song is mostly used and not calls so it would not be very relevant.

Within bird vocalizations, bird song is a specific group of vocalizations related to reproduction (territoriality and mating). The term bird calls describes a much broader group of vocalizations that include many contexts such as social cohesion, bonding, alarm, feeding, fighting... Therefore, these two concepts cannot be considered as equals, especially if the specific calls are not identified and described during the surveys. Changes in call activity throughout the breeding seasons may or may not be related to territory or mate selection at all but rather to any of the other possible context in which calling occurs (also likely to change during the season). Furthermore it is not mentioned any criteria whatsoever to consider a vocalization as a call or as a song and this is relevant especially if the most universal criterion is not used (e.g. songs are vocalizations related to mate attraction and territoriality). Other possible criteria would be the "acoustic complexity" or the fact that is produced mainly by one sex, which is also not mentioned. I did only a quick search but I found little information about Dupont's lark calls in the papers cited as I only could find the description of the territorial calls. For instance, if it happens to be that the so called territorial call is used also for mate attraction as claimed in the discussion (line 238), what would make this vocalization different from a song?

In line 217 it is considered that "calls" is one kind of vocalization and this is fundamentally wrong for many species since the term "bird calls" includes many kinds of vocalizations produced in different contexts.

Experimental design

Singing is "assumed" to be exclusive for males although it is becoming more obvious that females sing more often than thought. But, as far as I know, calling behavior is definitely not exclusive to any sex so, how can you be sure that calling activity is produced by the male only identifying the direction of the sound? This may bias or introduce confounding factors in the data and therefore the drawn conclusions because
1) there might be pseudoreplication and/or
2) calling activity could change for one of the sexes and not for the other which would lead to similar results as shown. The latter phenomenon has been reported in other species (e.g. dippers).

Statistical methods are very poorly reported. No distribution shown for response variable. Also, what family distribution and link function is used in the GLMM? This is basic information when reporting generalized linear mixed models. Finally it is crucial to show the procedure of building the model, running diagnostic test to ensure model assumptions are met and the selection procedure of the final model (or averaging). Model selection is especially important in observational studies.

Validity of the findings

Finding that song and calling activity differ throughout the season is probably expected and probably trivial by itself. Need some stronger justification for the study and a broader perspective in the discussion.

---

## Round 0.2 · accepted · Accept

· Academic Editor

Accept

I read the rebuttal and I believe that you have done a good work addressing the most important points of the reviews. Reviewer 2 is an expert in the field and also checked the new version of your paper, suggesting that it can be accepted for publication.

Reviewer 2 ·

Basic reporting

The presentation is much improved over the last submission. The writing is now quite clear and grammatical.

Experimental design

The authors answered my questions about their methods quite adequately, and I am now clear on what they did. I'm not convinced that they actually could tell accurately which birds were or were not within 250 meters, but I'm not terribly concerned about this point.

Validity of the findings

I am happy with how the authors present their conclusions in the present version

Additional comments

Thank you for your attention to my previous comments in revising.